# OpenReview forum: "Can SAEs reveal and mitigate racial biases of LLMs in healthcare?"
_ICLR.cc/2026/Conference — ICLR 2026 Poster_

### Official Review · Reviewer_yK5E · 2025-10-30

**Soundness:** 3
**Presentation:** 3
**Contribution:** 2
**Rating:** 4
**Confidence:** 5

**Summary:**

The paper aims to understand how LLMs encode patient race (white vs. Black) when prompted with clinical information, and how they use race when generating text about a patient. To do so, they identify SAE latent features in Gemma 2 models that predict race, and they show a few things with these SAE latents:
1. The latent that is most predictive of race = Black also activates on other words that suggest problematic associations learned by the LLM,
2. Steering via this latent (increasing its value and plugging the representation back into the model) leads to a higher rate of the model outputting problematic associations (e.g., belligerence),
3. Steering the latent reduces assignment of Black race to patient vignettes in conditions correlated with race (cocaine abuse, gestational hypertension, uterine fibroids),
4. Steering the latent does not substantially change the logits on whether the patient is recommended pain medication or not based on their clinical history.

**Strengths:**

The main strength of the paper is in attempting to understand how race is encoded in clinical settings - this is a laudable goal of interest to the community, since we don't want real applications of LLMs in healthcare to inadvertently cause harm. Coarse encodings of race are one way this can happen. So, the main strength of this paper is the choice to study this problem at the intersection of health, interpretability, fairness, etc.

The paper executes a fairly reasonable set of experiments, and they are explained clearly.

In terms of originality, I was most unsure about how much this paper adds to Ahsan et al. (2025) [1]. That paper was first posted in Feb 2025 (and will be presented at EMNLP soon). I hadn't seen this paper before, so I read it closely. After reading, here is my view on the originality of the present submission:
1. It uses SAEs, while [1] directly intervenes on the MLP activations.
2. The current paper has slightly more emphases on steering experiments, e.g. the pain medication experiment, while [1] only has one steering experiment on clinical vignette generation.

See further discussion under Weaknesses, because ultimately I am not convinced that this paper adds enough actionable insight to warrant acceptance. I'm giving a 4 to encourage the authors to try to improve this work, but I don't think it should be accepted in the current form.

[1] Ahsan et al: "Elucidating Mechanisms of Demographic Bias in LLMs for Healthcare" https://arxiv.org/pdf/2502.13319

**Weaknesses:**

There are three major weaknesses with this paper:
1. The tasks and evals are too toy. We can consider them one by one:
- Belligerence: Is this a type of task that people are doing (predicting whether a patient will be belligerent from their clinical notes)? If this task should matter on its own, there should be references; if the task should matter because it says something about other tasks, that link should be explicitly drawn out.
- Patient vignette generation: Why are we interested in this task? If it is for, e.g., generating a diversity of vignettes for clinical education, we should compare against an even simpler prompting baseline: "Please generate a vignette for a white patient." I know the Zack paper uses a similar setup but I believe it's an issue with that paper too.
- Clinical tasks (logit diff): Similarly, for this task, ignoring the fact that we do not see substantial differences, even if we did, we would need to show more specifically that the LLM is somehow using race to make *incorrect* predictions, not just that the predictions change.
2. Novelty compared to Ahsan et al. I would be willing to overlook this point if the paper found important new results by using SAEs (which is their main novelty). But, seeing as SAEs did not enable a new type of result or a new discovery of unexpected problematic associations, etc., the use of SAEs alone isn't sufficient novelty.
3. Taken together, (1) and (2) illustrate that the results of this paper aren't actionable. What is the failure mode here, and how this paper help solve it? More specifically, it is not surprising that there is a link in activation space between "Black" and "uterine fibroids", and this paper does not illustrate why this might cause harms in a real setting. This is an especially important question for this paper to answer, because the methodology or line of inquiry is not fundamentally new relative to Ahsan et al. (and possibly other papers I'm missing).

**Questions:**

See above.

---

> ### Author Response · Authors · 2025-11-17
> **Author response**
>
> We thank the reviewer for the review. Below is our response.
>
> **1. Too toy tasks**
>
> **Belligerence:** While the tasks considered here are not uniformly realistic, they are nonetheless grounded in the clinical domain and, we would argue, more meaningful than most of the evaluations used in the literature on SAEs. It is true that the probe for belligerence is not a realistic “task”, but it is illuminating. For example, we think it is problematic if an LLM associates Black individuals with increased risk of belligerence, even if this specific “prediction” is probably not of practical use—there are likely to be implications of such implicit associations for more practical clinical tasks. Prior work, for example, has shown that diagnostic accuracy of GPT 4.0 drops when input clinical cases are altered to contain patient disruptive behavior (irrelevant to the diagnosis) [1]. Having an implicit association of belligerence with a certain race can exacerbate health disparities [2]. Our analyses here establish that  the association does exist.
>
> [1 ] Schmidt, Henk G. et al. "Bias Sensitivity in Diagnostic Decision-Making: Comparing ChatGPT with Residents." Journal of General Internal Medicine 40.4 (2025)
>
> [2] Pierson, Emma, et al. "Using large language models to promote health equity." NEJM AI 2.2 (2025)
>
> **Vignette:** This task is used to elicit implicit biases internalized by the model. We agree it is not itself a practically useful task, but rather can be used as a lens to characterize implicit associations made by LLMs (as established by Zack et al.). In particular, vignette generation offers simplicity and limited confounding variables: The task involves a single condition and is not complicated by the presence of other information (unlike clinical notes). We wanted to first study if the associations we found in Section 2.2 can be intervened upon in the simplest set up (which we found is the case) and then proceed to more realistic tasks.
>
> **Clinical Tasks:** First, we do observe statistically significant differences between races. Second, these differences come about as a result of only intervening on the race via the corresponding SAE latent. Can the reviewer elaborate as to why they think this does not establish that the LLM is using race to inform predictions? Note that these are cases where race should not inform individual predictions.
>
> **Novelty compared to Ahsan et al.** Our work offers novel insights compared to Ahsan et al. primarily in that here we are examining SAEs as an interpretability tool rather than activation patching (which is what they studied). More specifically:
>
> i) They performed activation patching in vignette generation for conditions with biases *known beforehand* (they first reproduce results from Zack et al. and then localize associated demographics). By contrast, we investigate SAEs to reveal previously *unknown* biases that include stigmatizing associations (Section 2).
>
> ii) They show how implicit biases can be studied if race or gender are “patched” in. We investigate if reliance on race when explicitly mentioned can be detected and unlike their work, also show if it can be mitigated by using SAEs and intervening on appropriate latents.
>
> This is an analysis paper; the novelty is in our evaluation of SAEs for identifying and potentially mitigating bias in clinical tasks. SAEs are an increasingly popular interpretability tool, and we think this makes empirical evaluations of their utility important. There has been little work, to our knowledge, evaluating SAEs on clinical tasks; but then how is the community to know whether we should keep putting effort into SAEs or some other interpretability technique?
>
> Here we answered whether and to what degree SAEs might help for spotting and mitigating bias in clinical tasks, with mixed results. That does not mean the evaluation itself isn’t novel, only that the results are not uniformly positive for SAEs; this seems important for the community to know.
>
> **3. Unclear how associations cause harm:** Associations between Black individuals and uterine fibroids is neither surprising nor concerning in itself. Indeed, we state this (line 158) and do not box the condition in red in Figure 1 to suggest that it is problematic. The problem arises when such associations influence outputs for tasks that *do not* depend on race – this is what we study in Section 4.2. As stated on line 284 onwards, if the model does rely on race, we would like the model to explicitly state so (we want to use SAEs for transparency, not to enforce parity or prevent reliance on demographic attributes entirely), but we showed in Section 3 that this is not the case. In other words, the association in itself is not harmful, but implicit assumptions based on it for tasks that do not depend on race (such as our task, which requires strictly reasoning over the note) is not desirable.

---

> > ### Comment · Reviewer_yK5E · 2025-11-25
> > **Response to Authors**
> >
> > ### Responses
> >
> > *"For example, we think it is problematic if an LLM associates Black individuals with increased risk of belligerence, even if this specific “prediction” is probably not of practical use—there are likely to be implications of such implicit associations for more practical clinical tasks. Prior work, for example, has shown that diagnostic accuracy of GPT 4.0 drops when input clinical cases are altered to contain patient disruptive behavior (irrelevant to the diagnosis) [1]. Having an implicit association of belligerence with a certain race can exacerbate health disparities [2]. Our analyses here establish that the association does exist."*
> >
> > What you've shown is that there is a direction in the latent space of the model that fires on "Black", "preterm labor", "gunshot wounds", and a handful of other concepts (L153), and if you shift the latent space in this direction, it increases the chance the model will output "yes" to a stylized question about belligerence.
> >
> > This result does not directly imply anything about diagnostic accuracy; we would need to see a realistic experiment about diagnostic accuracy in order to say something about that.
> >
> > And, as someone who's very familiar with [2], I'm pretty confident that it does not say anything to support that "Having an implicit association of belligerence with a certain race can exacerbate health disparities".
> >
> > *"Second, these differences come about as a result of only intervening on the race via the corresponding SAE latent. Can the reviewer elaborate as to why they think this does not establish that the LLM is using race to inform predictions? Note that these are cases where race should not inform individual predictions."*
> >
> > A few things. First, just because this latent happens to correlate with race, it doesn't mean we can make a statement like "the LLM is using race". The latent is also correlated with preterm labor, so, for example, how do we know if the LLM is using preterm labor to make the prediction vs. using race? Second, the more important part of my comment was that "we would need to show more specifically that the LLM is somehow using race to make incorrect predictions, not just that the predictions change".
> >
> > ### Novelty
> >
> > The authors confirm that the primary value of the work is the SAE. However, I'm not convinced by this value:
> >
> > *"By contrast, we investigate SAEs to reveal previously unknown biases that include stigmatizing associations (Section 2)."*
> >
> > In Section 2, you mentioned that there is a direction in latent space that activates on both "Black" and reflects various social factors / determinants of health (incarceration, etc.). Is your claim that the SAE was necessary for this? Even if so, we return to the discussion above of why this is an important finding.
> >
> > *"SAEs are an increasingly popular interpretability tool, and we think this makes empirical evaluations of their utility important. There has been little work, to our knowledge, evaluating SAEs on clinical tasks; but then how is the community to know whether we should keep putting effort into SAEs or some other interpretability technique?"*
> >
> > This paper, unfortunately, doesn't seem to be an adequate evaluation of SAEs, so it doesn't shed much light on the question you pose. SAEs could certainly be very useful for clinical tasks (such as discovering differences in how doctors describe patients in clinical notes, which would be an example of what the aforementioned ref. [2] discusses), but the issue with the paper is with its choices of evaluation settings.
> >
> > ### Summary
> >
> > I think this paper is a good start, and some of the goals the authors describe are laudable - like studying if the model can faithfully explain whether race is contributing to its prediction or not.
> >
> > But in its current form, the paper is pretty scattered and therefore hard to draw any clear conclusions about (a) the shortcomings of LLMs in medical settings or (b) whether SAEs can help reveal bias in LLMs.
> >
> > I maintain my recommendation for rejection.

---

> ### Author Response · Authors · 2025-11-26
>
> We thank the reviewer for their engagement and thoughtful comments. We offer a few clarifications.
>
> **1. Response to**  *“What you've shown is that there is a direction in the latent space of the model that fires on "Black", "preterm labor", "gunshot wounds", and a handful of other concepts (L153), and if you shift the latent space in this direction, it increases the chance the model will output "yes" to a stylized question about belligerence. ...  how do we know if the LLM is using preterm labor to make the prediction vs. using race?.”*
>
> In fact, we also show in Tables 2 and 3 that intervening on this latent ***not only changes the response about belligerence but also the race of the patient***. Intervening on this particular latent alters predictions and *the stated race in the anticipated way.*
>
> The average activation strength on mentions of "African-American" is ~3x higher than that on associations in our corpus. We have revised Figure 1 and related description to better capture this. Figures 4 and 5 show examples in the general domain corpus as well.
>
> **2. Belligerence and [2]**
>
> We apologize for any confusion here. We are referring to the following statement in [2]: ***“Human biases contribute to inequity. For example, doctors describe Black patients as “difficult” more often than white patients, a textual indicator of the stigmatization and bias that can pervade the medical pipeline and contribute to health disparities.”***.
>
> Training LLMs on such biased data and then using them to make clinical decisions *is* one of the ways such biases can pervade the medical pipeline and contribute to health disparities. The notion of belligerence is one such stigmatizing concept and the Black latent shows such associations exist in LLMs.
>
> **3. Response to** *"we would need to show more specifically that the LLM is somehow using race to make incorrect predictions, not just that the predictions change"*
>
> In Table 6, we show statistically significant logit differences before any intervention. We restate that these differences are observed when we merely change patient race in the input; everything else remains the same. This in itself shows that race is affecting LLM output. We still believe that even a difference in predictions for tasks in which race should not affect output is worth investigating.
>
> We do wish to improve our work and would appreciate suggestions for evaluation tasks that more clearly show that an LLM is using race and making incorrect predictions.

---

### Official Review · Reviewer_tsQH · 2025-10-31

**Soundness:** 3
**Presentation:** 4
**Contribution:** 1
**Rating:** 4
**Confidence:** 3

**Summary:**

This paper uses Sparse Autoencoders (SAEs) to investigate racial bias in healthcare LLMs. SAEs successfully reveal problematic associations (e.g., Black patients being "belligerent") that Chain-of-Thought reasoning hides. However, SAE-based mitigation only works on simple tasks, failing in complex clinical scenarios.

**Strengths:**

* **Reveals Hidden Biases:** SAEs effectively identify problematic associations between race and stigmatizing concepts like "incarceration" and "cocaine."
* **Exposes Unfaithful CoT:** Demonstrates that Chain-of-Thought (CoT) explanations are unfaithful, hiding the model's reliance on race.
* **Domain-Specific Interpretability:** Highlights the necessity of re-interpreting SAE latents for the specific clinical domain.
* **Causal Steering:** Uses steering to causally confirm that activating the "Black latent" increases predictions of patient belligerence.

**Weaknesses:**

* **Weaker takeaways:** The authors are applauded for taking a very honest stance on their findings and avoiding making strong claims not supported by their findings. However, the paper's takeaways, especially for ICLR, seem kind of vague. Previous studies have applied SAEs to LLMs, and before reading studies, a reader would have expected a similar relevance if they were used for studying fairness. The only major finding seems to be showing SAE's outperforming COTs in "some" cases.
* **Other methods:** The paper seems to only compare SAEs to prompting-based methods of mitigation. It is not clear how other methods would compare.
* **Mitigation Fails:** The proposed mitigation (ablating race latents) fails to reduce bias in realistic, complex clinical tasks.
* **Limited Model Scope:** The analysis is limited to only the gemma-2 model family.
* **Limited Demographic Scope:** The study focuses only on "Black" and "white" racial categories from a single hospital dataset.

**Questions:**

See above, please.

---

> ### Author Response · Authors · 2025-11-17
> **Author response**
>
> We thank the reviewer for the review. Below is our response to the weaknesses.
>
> **1. Weaker takeaways:** SAEs have emerged as a popular interpretability tool, but just how useful they may be for high-stakes tasks remains an open question. We think it is important to try and evaluate SAEs in more realistic, domain-specific contexts: Hence the present effort. It is true that our results are mixed. SAEs do seem to offer one tool to spot reliance on sensitive patient features for some tasks (and more so than CoT), but their utility for steering is limited.
>
> We argue that it is important that as a community we do not only publish “positive” findings for interpretability methods; this risks overstating their usefulness. Our work offers a sober look at the performance of SAEs for a set of clinical tasks; our hope is that this motivates interpretability work moving forward.
>
> **2. Other methods:** Prompting seemed like the most straight-forward point of comparison (we also compare to CoT for interpretability). Perhaps the reviewer could suggest additional methods they would like us to compare with?
>
> **3. Mitigation Fails:** Yes, and we agree that this is disappointing. But given the growing interest in SAEs as an interpretability tool, we think it is important to highlight empirical limitations of this approach, especially in more realistic high-stakes settings. We feel the interpretability community at ICLR will benefit from knowledge of such shortcomings in complex clinical tasks—which are the sort of downstream applications where one might hope interpretability has the most to offer.
>
> **4. Limited Model Scope:** We are happy to report additional findings using OpenAI’s gpt-oss-20b. We used the open-source SAE corresponding to the middle layer (as we did with Gemma models) available on Neuronpedia and HuggingFace (https://huggingface.co/andyrdt/saes-gpt-oss-20b/tree/main/resid_post_layer_11/trainer_0). We found a latent that strongly activates on mentions of Black population but also activates on stigmatizing concepts similar to those in Gemma. Below are some examples (corresponding to Figure 1), where we enclose max-activating terms in brackets "[ ]".
>
> **[African-American]** Male sitting up in NAARD HENT...
>
> TAB 3 **[black]** female...
>
> Other son is currently **[incarcerated]**. She was…
>
> Remote history of **[cocaine]** abuse who presented in
>
> Previously in **[jail]** for armed robbery…
>
> PTSD, Polysubstance abuse (**[cocaine]** and marijuana)....
>
> This shows that our finding generalizes to other model families. We have included this result in the revised manuscript to demonstrate generalizability beyond gemma-2 model variants.
>
>
> **5. Limited Demographic Scope:** Our analysis is limited to the two races, as discussed in line 147, due to small sample sizes of other races. Specifically, prior work [1] has shown that using MIMIC data (the dataset we use here) may provide inaccurate estimates of disparities between other groups due to relatively small sample sizes.
>
> [1] Amir, Silvio, Jan-Willem van de Meent, and Byron C. Wallace. "On the impact of random seeds on the fairness of clinical classifiers." arXiv preprint arXiv:2104.06338 (2021).

---

### Official Review · Reviewer_xRkE · 2025-11-04

**Soundness:** 3
**Presentation:** 2
**Contribution:** 1
**Rating:** 4
**Confidence:** 3

**Summary:**

The authors tackle the problem of racial stereotypes in LLMs for clinical text. Specifically, they test whether sparse autoencoders can surface and control race-linked latents in clinical LLMs by identifying a "Black latent" in Gemma-2. The authors show that steering this latent with a patient belligerence prompt leads to a causal increase in "Yes" predictions, and reduced stereotyping in generated clinical vignettes, but ablating race latents do not improve downstream task biases significantly.

**Strengths:**

1. The authors tackle the problem of bias in clinical LLMs through a mechanistic interpretability perspective, which is an important real-world problem.

2. The authors conduct a fairly thorough set of tests to probe the "Black latent" that they discover.

**Weaknesses:**

1. The paper is essentially a case study on one specific type of bias (stereotypes against Black patients) in two specific open-source LLMs. It is unclear whether these findings would translate to biases against other demographic groups, or whether there would be a corresponding latent for all such biases. Further, it is unclear for what categories of demographics and clinical concepts the latents are disentangled.

2. I'm not convinced by the authors' argument in 4.2.1 that all of the biases probed in the paper are biases that we would actually want to eliminate. For example, gestational hypertension is more common in Black patients, and as long as race provides a signal in the real-world, I don't see any issue with predicting this disease with higher probability for Black patients. Intervening on this effect can actually worsen predictive accuracy for Black patients, which the authors should evaluate.

3. The authors study two relatively small open-source models. I would imagine both of these models have fairly poor performance on clinical prediction tasks, and so their real-world deployment is limited. The authors should report overall model performance and try a bigger model, e.g. Qwen 2.5 72B. It would also be interesting to see whether these findings hold for self-reasoning models.

4. The final conclusion of the paper, that SAEs can reveal racial associations but don't help with mitigation, is not very satisfying. The authors briefly speculate that entanglement may be the issue, but there are further investigations that can be done e.g. some of Anthropic's work on superposition, circuit tracing, etc, which would allow an explanation of why the discovered Black latent doesn't help with mitigation, and suggest other mitigation strategies.

**Questions:**

1. How can the authors be sure that the identified latent is a Black latent that spuriously relates with cocaine/gunshot/hypertension, instead of a hypertension latent that spuriously correlates with Black?

2. How does the Black latent relate with a potential White latent? Does one activating deactivate another? Is there a single race axis, or multiple distinct latents for each race?

---

> ### Author Response · Authors · 2025-11-17
> **Author response**
>
> We thank the reviewer for the review. Below is our response to the weaknesses and questions.
>
> **1. Generalization:** We focus on a concrete and important task using open-weight models that we believe are fairly representative of the state of the art. We argue that grounded case studies like this are important to inform the broader development and evaluation of interpretability methods. Any such focussed evaluation will necessarily trade off against generality; still, we think the broad adoption of LLMs in healthcare and documented history of racial bias in this domain warrants meaningful evaluation of interpretability methods in this context.
>
> **2. Not all biases need to be eliminated:** We apologize for the confusion. We in fact agree with the reviewer and offer further clarification here: In section 4.2 (line 275 onwards), we state explicitly that our goal is *not* to achieve demographic parity in all tasks — this is problematic in healthcare, as the reviewer rightly points out.  As discussed in the paper, if the model does rely on race, we would like the model to explicitly state so (we want *transparency*, not to enforce parity or prevent reliance on demographic attributes entirely), but we showed in Section 3 that this is not the case. In other words, while gestational hypertension *is* more common in Black patients (not due to genetics), the task we look at requires strictly reasoning over the note and retrieving clinically relevant evidence from an individual perspective; in this case making implicit assumptions based on population prevalence is not desirable.
>
> **3. Small models:** We believe that the models being modestly sized is a strength rather than a weakness. Hospitals and clinical research laboratories are resource constrained. They typically do not have access to large GPUs for on-premise deployment. Many hospitals are hesitant to use cloud-based frontier models given the sensitive nature of patient data and additional legal and administrative lift [1,4]. Works continue to investigate small-to-medium size model performance and have found them to be competitive for several types of clinical tasks  [2,3].
>
> [1] Riedemann, L., Labonne, M. & Gilbert, S. The path forward for large language models in medicine is open. npj Digit. Med. 7, 339 (2024). https://doi.org/10.1038/s41746-024-01344-w
>
> [2] Somani, Sulaiman, et al. "Performance Benchmarking of Smaller Language Models Against GPT-4 for Predicting Reasons for Oral Anticoagulation Nonprescription in Atrial Fibrillation." Circulation 152.Suppl_3 (2025): A4366575-A4366575.
>
> [3] Wu, Jiageng, Bowen Gu, Ren Zhou, Kevin Xie, Doug Snyder, Yixing Jiang, Valentina Carducci et al. "BRIDGE: Benchmarking Large Language Models for Understanding Real-world Clinical Practice Text." arXiv preprint arXiv:2504.19467 (2025).
>
> [4] Woo, Elizabeth Geena, et al. "Synthetic data distillation enables the extraction of clinical information at scale." npj Digital Medicine 8.1 (2025): 267.
>
> **4. Unsatisfying conclusion:** We agree with the reviewer that this is somewhat unsatisfying, but we also feel strongly that it is important that “negative” (or at least “mixed”) results for interpretability methods are published, else we risk as a community publishing only “positive” results for such techniques and exaggerating their effectiveness. As reviewer tsQH points out: We are aiming for honesty here, and our hope is that this work provides accurate empirical results concerning the use of SAEs (a currently popular set of methods) for spotting and potentially mitigating racial bias in healthcare, even if these results are not as good as one might hope.
>
> **Q1. Is this a Black latent?** We believe this is the case because activation patterns on the general-domain corpus also align with this interpretation. Figures 4 and 5 in the Appendix are screenshots from Neuronpedia. We see that the latent activates on mentions of Black population (“African” appears in the top logits for both models) and also on associated concepts such as, Morehouse College and Howard University, which are historically Black universities.  It seems unlikely to us that a “hypertension” latent would fire on such inputs. In the clinical space, the associated concepts translate to prevalent conditions such as gestational hypertension. We have added references to Figures 4 and 5 in Section 2.2. We have also modified Figure 1 and related description to explain this better.
>
> **Q2. Relation to other race latents** We found multiple latents that activate on mentions of any type of race. We also found latents, like the Black latent, that exclusively activate on a specific race. Table 10 contains descriptions of race related latents. A race-exclusive latent can be used to steer (as shown in Section 3) and depending on the magnitude, overwrite existing race information.

---

### Official Review · Reviewer_vqoq · 2025-11-07

**Soundness:** 4
**Presentation:** 4
**Contribution:** 3
**Rating:** 8
**Confidence:** 3

**Summary:**

The paper investigates whether sparse autoencoders (SAEs) can help interpret and reduce racial bias in large language models (LLMs) applied in healthcare settings with a use case of the MIMIC dataset. They apply SAEs  to identify latent units that correlate with race (specifically Black vs White patients) when processing discharge summaries. They find a “Black latent” (for two variants of a model, gemma-2-2B-it and gemma-2-9B-it) that activates not only on mentions of “African American” but also on stigmatizing concepts (incarceration, gun-shots, cocaine use) in the clinical text. They perform steering experiments: by intervening on the latent (increasing its activation) they show that model outputs shift such that the model assigns a higher “risk of becoming belligerent” when the patient is steered to be “more Black”. They test whether ablating the race-latent (or a set of race-related latents) reduces bias in downstream tasks: (1) a vignette generation task and (2) more realistic clinical prediction tasks (risk prediction, pain management). They find that in the controlled vignette generation setting the SAE intervention reduces bias more than prompting; but in realistic tasks, the mitigation effect is very small.

**Strengths:**

I really like the use of race-correlated latents to investigate bias and also appreciate that the authors show that CoT is not as useful in this regard.
I also appreciate the steering experiments, which shows some notion of causality here.

**Weaknesses:**

-One major limitation is using just two models from the same family. In addition, I am curious why they used the Gemma family instead of the medgemma family, which specifically was trained for medical tasks.
-It would be nice to show some examples of the CoT which failed to catch the bias in the appendix
-While there is an ethics section, emphasize how latent‐steering tools could be misused (e.g., malicious “race injection” in model inputs) and how to guard against that.
-Some of the effect sizes are very small; possible to bootstrap and include confidence intervals?

**Questions:**

Could this also be tested on other open source models?

---

> ### Author Response · Authors · 2025-11-17
> **Author response**
>
> We thank the reviewer for the review and respond below.
>
> **1. Models from the same family:** Models from the same family: We are happy to report additional findings using OpenAI’s gpt-oss-20b (https://huggingface.co/openai/gpt-oss-20b) We used the open-source SAE corresponding to the middle layer (as we did with Gemma models) available on Neuronpedia and HuggingFace (https://huggingface.co/andyrdt/saes-gpt-oss-20b/tree/main/resid_post_layer_11/trainer_0). We found a latent that strongly activates on mentions of Black population but also activates on stigmatizing concepts similar to those in Gemma. Below are some examples (corresponding to Figure 1), where we enclose max-activating terms in brackets "[ ]".
>
> **[African-American]** Male sitting up in NAARD HENT
>
> TAB 3 **[black]** female
>
> Other son is currently **[incarcerated]**. She was…
>
> Remote history of **[cocaine]** abuse who presented in
>
> Previously in **[jail]** for armed robbery…
>
> PTSD, Polysubstance abuse (**[cocaine]** and marijuana)....
>
> This shows that our finding generalizes to other model families. We have included this result in the revised manuscript to demonstrate generalizability beyond gemma-2 model variants.
>
> **2. Gemma instead of MedGemma:** Our decision to use Gemma instead of MedGemma is based on several prior works that have shown that domain-adapted models show limited improvements over their base models over a range of clinical tasks [1, 2].
>
> [1] Jeong, Daniel P., et al. "The limited impact of medical adaptation of large language and vision-language models." arXiv preprint arXiv:2411.08870 (2024).
>
> [2] Ceballos-Arroyo, Alberto Mario, et al. "Open (clinical) llms are sensitive to instruction phrasings." Proceedings of the 23rd Workshop on Biomedical Natural Language Processing. 2024
>
> **3. Other** The reviewer correctly points out (as we also state in line 360) that the effect size of race latents is small.  We will revise Figure 2 plots and include confidence intervals. We will also add additional examples in the Appendix for failed CoT. We will elaborate in the ethics section about misuse of and potential guardrails again harmful steering with race latents.

---

### Official Review · Reviewer_vqoq · 2025-11-07

**Soundness:** 4
**Presentation:** 4
**Contribution:** 3
**Rating:** 8
**Confidence:** 3

**Summary:**

The paper investigates whether sparse autoencoders (SAEs) can help interpret and reduce racial bias in large language models (LLMs) applied in healthcare settings with a use case of the MIMIC dataset. They apply SAEs  to identify latent units that correlate with race (specifically Black vs White patients) when processing discharge summaries. They find a “Black latent” (for two variants of a model, gemma-2-2B-it and gemma-2-9B-it) that activates not only on mentions of “African American” but also on stigmatizing concepts (incarceration, gun-shots, cocaine use) in the clinical text. They perform steering experiments: by intervening on the latent (increasing its activation) they show that model outputs shift such that the model assigns a higher “risk of becoming belligerent” when the patient is steered to be “more Black”. They test whether ablating the race-latent (or a set of race-related latents) reduces bias in downstream tasks: (1) a vignette generation task and (2) more realistic clinical prediction tasks (risk prediction, pain management). They find that in the controlled vignette generation setting the SAE intervention reduces bias more than prompting; but in realistic tasks, the mitigation effect is very small.

**Strengths:**

I really like the use of race-correlated latents to investigate bias and also appreciate that the authors show that CoT is not as useful in this regard.
I also appreciate the steering experiments, which shows some notion of causality here.

**Weaknesses:**

-One major limitation is using just two models from the same family. In addition, I am curious why they used the Gemma family instead of the medgemma family, which specifically was trained for medical tasks.
-It would be nice to show some examples of the CoT which failed to catch the bias in the appendix
-While there is an ethics section, emphasize how latent‐steering tools could be misused (e.g., malicious “race injection” in model inputs) and how to guard against that.
-Some of the effect sizes are very small; possible to bootstrap and include confidence intervals?

**Questions:**

Could this also be tested on other open source models?

---

> ### Author Response · Authors · 2025-11-17
> **Author response**
>
> We thank the reviewer for the review and respond below.
>
> **1. Models from the same family:** Models from the same family: We are happy to report additional findings using OpenAI’s gpt-oss-20b (https://huggingface.co/openai/gpt-oss-20b) We used the open-source SAE corresponding to the middle layer (as we did with Gemma models) available on Neuronpedia and HuggingFace (https://huggingface.co/andyrdt/saes-gpt-oss-20b/tree/main/resid_post_layer_11/trainer_0). We found a latent that strongly activates on mentions of Black population but also activates on stigmatizing concepts similar to those in Gemma. Below are some examples (corresponding to Figure 1), where we enclose max-activating terms in brackets "[ ]".
>
> **[African-American]** Male sitting up in NAARD HENT
>
> TAB 3 **[black]** female
>
> Other son is currently **[incarcerated]**. She was…
>
> Remote history of **[cocaine]** abuse who presented in
>
> Previously in **[jail]** for armed robbery…
>
> PTSD, Polysubstance abuse (**[cocaine]** and marijuana)....
>
> This shows that our finding generalizes to other model families. We have included this result in the revised manuscript to demonstrate generalizability beyond gemma-2 model variants.
>
> **2. Gemma instead of MedGemma:** Our decision to use Gemma instead of MedGemma is based on several prior works that have shown that domain-adapted models show limited improvements over their base models over a range of clinical tasks [1, 2].
>
> [1] Jeong, Daniel P., et al. "The limited impact of medical adaptation of large language and vision-language models." arXiv preprint arXiv:2411.08870 (2024).
>
> [2] Ceballos-Arroyo, Alberto Mario, et al. "Open (clinical) llms are sensitive to instruction phrasings." Proceedings of the 23rd Workshop on Biomedical Natural Language Processing. 2024
>
> **3. Other** The reviewer correctly points out (as we also state in line 360) that the effect size of race latents is small.  We will revise Figure 2 plots and include confidence intervals. We will also add additional examples in the Appendix for failed CoT. We will elaborate in the ethics section about misuse of and potential guardrails again harmful steering with race latents.

---

### Author Response · Authors · 2025-11-24
**General author response**

We thank the reviewers for their thoughtful feedback. We are happy the reviewers found our work to be thorough and honest in its analysis of SAEs in a realistic and high-stakes context.

We also appreciate the areas of improvement suggested by the reviewers and offer our overall response here.

**1. Limited Model Scope:** We are happy to report additional findings using OpenAI’s gpt-oss-20b. We used the open-source SAE corresponding to the middle layer (as we did with Gemma models) available on Neuronpedia and HuggingFace (https://huggingface.co/andyrdt/saes-gpt-oss-20b/tree/main/resid_post_layer_11/trainer_0). We found a latent that strongly activates on mentions of Black population but also activates on stigmatizing concepts similar to those in Gemma. Below are some examples (corresponding to Figure 1), where we enclose max-activating terms in brackets “[]”.

**[African-American]** Male sitting up in NAARD HENT

TAB 3 **[black]** female

Other son is currently **[incarcerated]**. She was…

Remote history of **[cocaine]** abuse who presented in
Previously in **[jail]** for armed robbery…

PTSD, Polysubstance abuse (**[cocaine]** and marijuana)....

***This shows that our finding generalizes to other model families and bigger models***. We have included this result in the revised manuscript (section 2.2 and Appendix B.1) to demonstrate generalizability beyond gemma-2 model variants.

**2. Unsatisfying conclusion (results not uniformly “positive” for SAEs):** We feel strongly that it is important to publish negative or mixed results for interpretability techniques. Our work provides an honest and thorough analysis of SAEs, a currently popular interpretability method, for spotting and potentially mitigating racial bias in healthcare. If we only allow uniformly “positive” results for interpretability techniques to be published, we risk exaggerating the utility of such methods. Our work highlights what SAEs can help with in this space (such as revealing previously unknown biases) and where they are less successful. This can inform work on SAEs going forward.

**3. Generalization to other racial groups:** We focus on Black and white patients owing to the limitations of data (MIMIC, the dataset we use, is the largest publicly available repository of electronic health records). Prior work (Amir et al., 2021) has shown that using such data may provide inaccurate estimates of disparities between other groups due to relatively small sample sizes.

**4. (Un)problematic bias:** In section 4.2 (line 275 onwards), we state explicitly that our goal is *not* to achieve demographic parity in all tasks — this is problematic in healthcare. Associations between Black individuals and clinical conditions is neither surprising nor concerning in itself. We state this (line 158) and do not box the conditions in red in Figure 1 to suggest that it is problematic (we have also enhanced the figure title to make this clearer).  The problem arises when such associations influence outputs for tasks that do not depend on race – this is what we study in Section 4.2. As stated on line 292 onwards, if the model does rely on race, we would like the model to explicitly state so (we want to use SAEs for transparency, not to enforce parity or prevent reliance on demographic attributes entirely), but we showed in Section 3 that this is not the case. In other words, the associations as such are not harmful, but implicit assumptions based on them for tasks that do not depend on race is not desirable. The tasks we look at require retrieving clinically relevant evidence from an individual perspective; in this case making implicit assumptions based on population prevalence is not desirable.

---

### Meta-Review · Area_Chair_tQju · 2025-12-05

**Summary:**

The reviewers had different concerns with this paper, while recognizing that its goal was valuable and that experiments were overall sound and well presented:
- use of a single LLM model family of smaller size
- inconclusive results when it comes to SAEs for bias mitigation
- correlation vs causation questions when it comes to the signals embedded in the detected racial latents, and their effects
- evaluations limited to one dataset and 2 racial groups (Black vs White)

**Reviewer Concerns:**

The authors have provided careful responses to each reviewer. I believe that the addition of another model was necessary, but maybe not the main concern shared across reviewers.

I do share some concerns about the outcomes of the paper when it comes to SAEs and whether other mitigation strategies could have been explored. I however support the publication of mixed results to encourage more nuance in the community.

I would have liked to see other biases or datasets be explored. The main justification for using only 2 groups is related to the use of MIMIC, but nothing prevents other datasets to be used and the scope of the paper in terms of its analysis is somewhat limited.

**Reviewer Scores:**

One reviewer suggested acceptance and would likely maintained their score after rebuttal.

The other 3 reviewers have scored the paper a 4, with one reviewer explicitly maintaining their score after rebuttal. It is unclear whether the other 2 reviewers would have amended their score, but I believe it is possible that the addition of another model and the provided justifications for some choices would have obtained a slightly more positive outcome.

As the scores for presentation and soundness were excellent and the scores on contribution were mixed, I would like to support the paper for publication. I believe that publishing an analysis paper is more difficult than publishing a technical contribution and the message in this paper seems valuable to the community (even if reinforcing prior work). I would however encourage the authors to ensure that the language in the paper correctly reflects that the latents identified only reflect correlations and not causation, and to consider all the reviewers' feedback before publication.

---

### Decision · Program_Chairs · 2026-01-26

Accept (Poster)